

# The effects of rain and evapotranspiration statistics on groundwater recharge estimations under semi-arid environments

Tuvia Turkeltaub[1] and Golan Bel[2]

[1]Zuckerberg Institute for Water Research, Blaustein Institutes for Desert Research, Ben-Gurion University of the Negev, Sede Boqer Campus 8499000, Israel
[2]Department of Solar Energy and Environmental Physics, Blaustein Institutes for Desert Research, Ben-Gurion University of the Negev, Sede Boqer Campus 8499000, Israel

**Correspondence:** Tuvia Turkeltaub (tuviat@bgu.ac.il)

**Abstract.** Better understanding the effects of rainfall and evapotranspiration statistics on groundwater recharge requires long time series of these climate variables. However, long records of the relevant variables are scarce. To overcome this limitation, time series of the rainfall and evapotranspiration are often being synthesized using different methods. Here, we attempt to study the dependence of estimated groundwater recharge on the synthesis methods. We focus on regions with semi-arid climate con-

ditions and soil types. For this purpose, we used long records of climate data that were measured in two semi-arid locations with similar seasonal and annual potential evapotranspiration rates but different seasonal rain distributions. Stochastic daily rain and potential evapotranspiration time series were synthesized according to the monthly empirical distributions. This suggested synthesis method preserves only certain aspects of the measured statistics. Therefore, different correction methods were considered to match the synthesized time series to the measured annual or monthly statistics. Groundwater recharge fluxes

were calculated using the 1D Richards equation, for several typical semi-arid soil types, and by prescribing the synthesized rain and potential evapotranspiration as atmospheric conditions. The estimated groundwater recharge fluxes are sensitive to the synthesis method. However, the ratio between the groundwater recharge and the total rain does not show the same sensitivity. The effects of the synthesis methods are shown to be independent of the data used as observations. These findings suggest that assessment of groundwater recharge under current and future climate conditions depend on the synthesis method used for rain

and evapotranspiration.

## 1 Introduction

Rainfall characteristics such as total amount, intensity, and frequency, are important climate factors that strongly influence groundwater recharge (GR) rates (Small, 2005; Nasta et al., 2016; Barron et al., 2012). Various studies have illustrated that rain intensity is superior to the total annual rainfall in determining GR (e.g. Ng et al., 2010; Owor et al., 2009). Previous

studies in semi-arid areas illustrated that coarser soils facilitate higher recharge rates than finer soils (Keese et al., 2005; Wohling et al., 2012; Crosbie et al., 2013). These studies presented statistical relationships between annual rain and GR for each one of the reported soil types (sand, loam, and clay loam) or according to clay content. However, it has been demonstrated that using annual rainfall may yield large uncertainties in recharge estimations due to the seasonal effects (Moeck et al.,





2020; Small, 2005; Cuthbert et al., 2019). Small (2005) illustrated that, in sandy and sandy loam soils, seasonal rainfall

variations have a stronger effect on recharge than storm size distribution. An additional factor that may impede establishing a

relationship between rain and recharge is the thickness of the unsaturated zone. Many semi-arid and arid areas across the world

are characterized by a thick unsaturated zone ($> 15$ m), where water travel times can be on the order of years or decades (Fan

et al., 2013; Gurdak et al., 2007; Cao et al., 2016; Turkeltaub et al., 2015, 2018; Scanlon et al., 2009; Moeck et al., 2020; Cao

et al., 2016; Gurdak et al., 2007). Current historical rain records are not long enough to enable a complete characterization of

the relationship between rain characteristics (number of rainy days, storm duration, rain intensity, etc.) and GR in semi-arid and

arid areas. In order to overcome this limitation, assuming that the rain characteristics do not change over the period of interest,

long rain time series can be generated (synthesized) from the statistics of the measured rain (Small, 2005; Rodriguez-Iturbe

et al., 1999).

To generate synthetic rain, the statistics of both the rain occurrences and the rain intensities must be defined (Tencaliec et al.,

2020). The number of rain events is a discrete variable, while the amount of rain in each event is a continuous one. Thus,

different models/distributions are implemented to describe each of the components yielding the rain series (Tencaliec et al.,

2020). Previous studies that analyzed the relationship between rain statistics and GR used the Poisson approach to generate

synthetic rain (Small, 2005; Rodriguez-Iturbe et al., 1999; Burton et al., 2008).

In general, an exponential distribution is used to describe the distribution of dry intervals between sequential rainfall events,

the storm depth (total amount of rain in the storm), and the rain rate (Small, 2005). In order to preserve the seasonal or annual

means, the parameters of the different distributions (i.e., the distributions of the dry/wet intervals and rain intensity) are assumed

to be dependent such that the means of interest are conserved (Istanbulluoglu and Bras, 2006). Note that this implies replacing

correlated variables with independent ones that include parameters that are not derived from the measurements. To include the

seasonal rain distribution, the parameters of the exponential distributions must be defined for each month or by simply dividing

the year into two seasons, dry and wet (Small, 2005; Snyder et al., 2003). Other studies suggested using a nonstationary Markov

chain or a Bernoulli distribution to describe the occurrence of rain and gamma distributions to describe the rainfall amounts

(Stern and Coe, 1984; Lima et al., 2021). These methods suffer from an over-parameterization or an underestimation of rain

amounts (Tencaliec et al., 2020; Snyder et al., 2003).

While the effect of rain statistics on GR has received much attention, knowledge concerning the impact of the potential

evapotranspiration (ETref) statistics on GR fluxes is limited. It was previously demonstrated that GR may decrease despite

an increase in the total annual rain (Rosenberg et al., 1999; Kingston and Taylor, 2010). This was attributed to increases in

monthly temperature, which in turn, increases the monthly potential and actual ET rates. Small (2005) generated ETref time

series by accounting for the ETref mean annual value and seasonal amplitude. It was found that the synchronization (or lack of

it) dictates the GR amounts. However, previous studies have shown that reliance on monthly or annual means of meteorological

variables may lead to bias in GR predictions (Wang et al., 2009; Batalha et al., 2018).

The current study aims to identify the important characteristics of the local rain and ETref for estimating the diffuse recharge

under semi-arid climate conditions. Relatively long rain and Penman-Monteith ETref records ($> 40$ years), in two different

locations with different seasonal rainfall patterns but comparable ETref rates, to generate synthetic daily rain and ETref time





series. These time series preserve the daily statistics but not necessarily the other characteristics of the rain and ETref. To

overcome this issue, several new methods, preserving different characteristics of the measured rainfall and ETref records, are applied, and the different synthetic data series were used to assign the atmospheric boundary conditions for GR simulations. Recharge fluxes are simulated by solving the Richards equation with different hydraulic parameters corresponding to typical semi-arid soil types. These soil parameters were selected according to the global distribution of the soil texture in semi-arid and arid environments. Ultimately, the effects of the seasonal rain distribution and the synthesis methods on the estimated GR

fluxes are discussed. Note that we focus on the simpler case of bare and homogeneous soil. Estimations of actual GR fluxes in the presence of vegetation and preferential flow require specific field details and are beyond the scope of the current study.

## 2 Methods

### 2.1 Climate data

The meteorological datasets used in this study were obtained at the Beit Dagan ($32°00'$N, $34°49'$E, 30 m AMSL) and Shenmu

($38°55'$N, $110°7'$E, 926 m AMSL) Meteorological Stations. These stations were selected for their similar seasonal and annual Penman reference evapotranspiration (ETref) (Allen et al., 1998) rates. However, their annual and seasonal rainfall distributions are different (Figure 1).

According to the meteorological climate records, the Shenmu site is characterized as a semi-arid temperate location with a mean, maximum, and minimum annual temperature of 9, 16, and 3 ($°C$), respectively. The mean annual precipitation is 420

mm with approximately 70% of it falling between June and September (in summer, mostly monsoon rain). This implies that the rainy season corresponds to the months with the largest ETref rates. Maximum and minimum ETref rates occur during June (5.4 mm/day) and December (0.6 mm/day), respectively. Beit Dagan represents a Mediterranean climate, where winter is the rainy season (October to March), implying that the rainy season corresponds to the period with the lowest ETref rates. The mean, maximum, and minimum annual temperatures are 19, 31, and 7 ($°C$), respectively. The mean annual precipitation is 553

mm/year. Maximum and minimum ETref rates occur during July (5.9 mm/day) and January (1.8 mm/day), respectively.

The dataset for Beit Dagan spans 57 years of records (1964–2021), and the dataset for Shenmu spans 54 years of record (1961–2014). Both records are considered long enough to represent the natural climate variability (Döll and Fiedler, 2008). The climate datasets contain daily measurements of mean air temperature ($°C$), maximum temperature ($°C$), minimum temperature ($°C$), precipitation (mm), relative humidity (%), sunshine hours (hr), and wind speed (m/s). Using these datasets, the daily

ETref rates were calculated according to the Penman-Monteith equation (Allen et al., 1998) for each meteorological station (see Figure 1 for the precipitation and ETref data). Note that 3237 days of the ETref in Beit Dagan are missing, where the longest gap is between 22/03/2008 and 31/12/2014. The data were used to derive the empirical probability distribution function (ePDF) of the variables. Therefore, we did not need to extrapolate the missing values, and the ePDFs are based only on the existing data.





In addition to the measured data we also used the CRU TS 3.2 data set (Harris et al., 2014) that was downscaled to daily values with ERA40 (1958–1978, Uppala et al. (2005)) and ERA-Interim (1979–2015, (Dee et al., 2011)). The downscaling method is described more extensively in van Beek (2008).

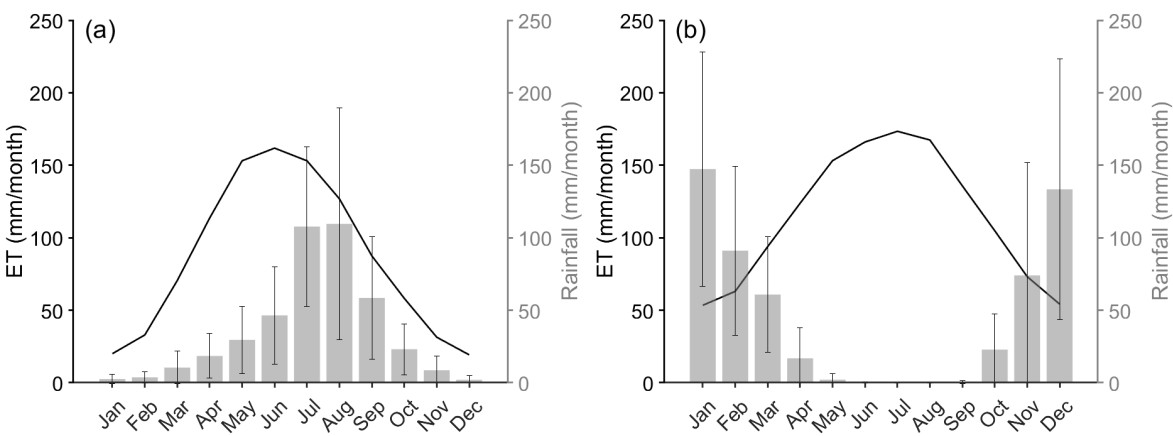

**Figure 1.** Climate data from meteorological stations located in (a) Shenmu, Loess Plateau, China and (b) Beit-Dagan, Israel. The climate data are presented as mean and STD perennial monthly values (the original data were measured at a daily resolution).

## 2.2    Generation of rain and ETref time series

The measured rain (MR) records were analyzed for each calendar month separately. For each month, we derived the distribution

of the number of rainy days and the distribution of the daily rain amount. The synthetic rain was generated for each month according to these two characteristics. First, the number of rainy days was drawn, obviously an integer number, and then for each day, the amount of rain was assigned (a number drawn from a continuous probability distribution). Using this method to generate the synthetic rain, we conserve the daily rain statistics (hereafter denoted as DS). In order to assess the importance of the distribution of the number of rainy days in each month, we also generated synthetic rain time series where we used the

average number of rainy days for each month, i.e., a fixed number of rainy days (the synthetic rain generated using this method is hereafter denoted as FNRD). The amount of rain in each of these days was drawn from the corresponding distribution of the daily rain amount. In order to illustrate the importance of the fact that the rain is limited to a certain number of days, we also considered a scenario in which the total monthly rain, generated using the DS method, was spread equally over all the days of the month. This scenario is hereafter denoted as UDDS.





However, the above procedures do not account for the correlation between the number of rainy days and the total amount of rain in the month nor for the correlations between the rain amounts within successive months. Therefore, we adopted several methods to correct the daily rain amounts as detailed below. The first correction method aimed at correcting the daily rain amounts such that the monthly averages and standard deviations (STDs) of the synthetic rain and the measured rain are the same for each calendar month. This method is denoted as monthly corrected daily statistics (MCDS). Practically, the correction

was applied by considering the DS rain series and multiplying each value in this series by the ratio between the measured STD of the total monthly rain and the STD of the total monthly rain of the synthesized DS series for the corresponding calendar month. Next, we added to each value in the synthetic series the difference between the measured and synthetic monthly mean divided by the number of rainy days in that month. Note that this correction could possibly have resulted in negative rain amounts; therefore, we only considered the rainy days with amounts above the correction in order to ensure positive values of

rain amounts and to conserve the number of rainy days. Mathematically, this correction method is described as:

$$\text{MCDS}(y,m,d) = \text{DS}(y,m,d)\frac{\tilde{\sigma}_M(m)}{\tilde{\sigma}_{DS}(m)} + \frac{\tilde{\Delta}(m)}{\text{Nm}(y,m)}. \tag{1}$$

Here, $\text{MCDS}(y,m,d)$ is the amount of rain in year $y$, month $m$, and day $d$ of the corrected synthetic rain series. $\text{DS}(y,m,d)$ is the amount of rain in year $y$, month $m$, and day $d$ of the DS rain series. $\tilde{\Delta}(m) \equiv \langle\text{MR}(m)\rangle - \langle\text{DS}(m)\rangle$ is the difference between the measured and the DS monthly rain averages for month $m$. The averages are defined as:

$$\langle\text{MR}(m)\rangle = \frac{1}{N_y}\sum_{1}^{N_y}\sum_{d\in m}\text{MR}(y,m,d), \tag{2}$$

where $N_y$ is the number of years spanned by the record. The average synthetic monthly rain is defined similarly. $\text{Nm}(m)$ is the number of rainy days in month $m$ of year $y$ with a rain amount large enough such that $\text{MCDS}(y,m,d) > 1$ if the correction is applied. If this is not the case, the amount of rain remains equal to its value in the DS series. $\tilde{\sigma}_{DS}(m)$ and $\tilde{\sigma}_M(m)$ are the STDs of the total monthly rain for the DS series and the measured rain of the calendar month $m$, respectively. The STD of the

measured monthly rain is:

$$\tilde{\sigma}_M(m) = \sqrt{\frac{1}{N_y - 1}\sum_{1}^{N_y}\left(\sum_{d\in m}\text{MR}(y,m,d) - \langle\text{MR}(m)\rangle\right)^2}. \tag{3}$$

In the second method, we applied a similar correction to the DS series, but rather than conserving the monthly mean and STD, we conserved the annual mean and STD. Mathematically, this correction may be written as:

$$\text{ACDS}(y,m,d) = \text{DS}(y,m,d)\frac{\sigma_M(y)}{\sigma_{DS}(y)} + \frac{\Delta(y)}{\text{Na}(y)}. \tag{4}$$

$\Delta(m) \equiv \langle\text{MR}\rangle_a - \langle\text{DS}\rangle_a$ is the difference between the measured and the DS annual rain averages. The annual averages are defined as:

$$\langle\text{MR}\rangle_a = \frac{1}{N_y}\sum_{1}^{N_y}\sum_{d\in m}\sum_{m\in y}\text{MR}(y,m,d), \tag{5}$$





and similarly for the DS series. The STD of the measured annual rain is:

$$\sigma_M = \sqrt{\frac{1}{N_y - 1} \sum_1^{N_y} \left( \sum_{d \in m} \sum_{m \in y} \mathrm{MR}(y,m,d) - \langle \mathrm{MR} \rangle_a \right)^2}, \tag{6}$$

and similarly for the DS series. It is important to note that in both the MCDS and ACDS corrections, we ensured that in each rainy day, there is at least 1 mm of rain, and we conserved the rainy days (namely, each rainy day in the DS series is also a rainy day in the MCDS and ACDS series).

Three methods, similar to those described above for the rain synthesis, are used to establish the synthetic ETref time series. The first method is similar to the DS method, where empirical probability density functions were established from the daily ETref records for each calendar month separately. Subsequently, the synthetic ETref time series were created by random sampling of ETref values from the corresponding empirical distributions. This method is hereafter denoted as ETDS. The other two methods involve corrections of the ETDS series. The ETDS time series were corrected to monthly (ETDSMC) and yearly (ETDSAC) statistics, similarly to the abovementioned procedure (see equations (1) - (6)). Additional ETref synthesis methods involve establishing the empirical density functions of ETref, for each calendar month, for rainy and dry days separately. The ETref values in the synthetic series are sampled from the corresponding empirical distribution according to rainy and dry days in the synthetic rain series. Namely, if the synthetic rain series shows rain on the specific day, the synthetic ETref value is randomly sampled from the empirical distribution of ETref values for rainy days in the corresponding calendar month. If the synthetic rain series shows no rain on that day, the synthetic ETref value is randomly sampled from the empirical distribution of ETref values for dry days in the corresponding calendar month. This method is hereafter denoted as ETWD. Here as well, the ETWD time series were corrected to conserve the monthly (ETWDMC) and annual (ETWDAC) statistics of ETref. The last ETref synthesis method is mostly used as a reference to demonstrate the importance of the daily values' statistics. In this method, the daily values for each calendar month, throughout the entire record for each station, were averaged, and ETref is assumed to be constant (equal to the average value for the corresponding calendar month) for each calendar month. This method is hereafter denoted as ETUD.

In the supporting information, comparisons between the cumulative distribution functions (CDFs) of the measured and synthesized daily rain, the number of rainy days, and the daily ETref values in each calendar month are presented. The two-sample Kolmogorov-Smirnov test indicated that the synthesized and the measured distributions are statistically similar for the DS, ETDS, and ETWD methods. Implementing the different methods allowed us to examine the sensitivity of the GR to the different statistical characteristics of the rain and the ETref. For convince, Table 1 provides a detailed list of all the abbreviations used for the different rain and ETref synthesis methods.

For each method, we used 50 different realizations of the synthetic rain series. The variance of the estimated GR includes two components: the temporal variability within each realization of the atmospheric conditions (rain and ETref) and the variability between different realizations. The total variability of the annual GR is defined as:

$$\sigma_{total}^2 = \frac{1}{N_r} \frac{1}{N_y} \sum_{y=1}^{N_y} \sum_{r=1}^{N_r} (\mathrm{GR}(r,y) - \langle \mathrm{GR} \rangle)^2, \tag{7}$$





**Table 1.** Description for the abbreviations of the different rain and ETref methods

| Abbreviations | Description |
| --- | --- |
| DS | Synthetic rain that is generated by conserving the measured daily rain statistics for each calendar month (i.e., number of rain days and the daily rain amount.) |
| FNRD | Synthetic rain that is generated using a fixed number of rainy days (i.e., the number of rainy days is set equal to the average for each calendar month) but preserving the daily rain amount statistics for each month. |
| UDDS | Rain and ETref are spread equally over all the days of the month, preserving the total monthly values statistics. |
| MCDS | Correcting the daily rain amounts of the DS time series such that the monthly averages and STDs of the synthetic rain match the measured statistics for each calendar month. |
| ACDS | Correcting the daily rain amounts of the DS such that the annual average and STD of the synthetic rain match the measured statistics. |
| ETDS | Synthetic ETref that is generated by conserving the measured daily ETref statistics for each calendar month. |
| ETDSMC | Correcting the daily ETref values of the ETDS time series such that the monthly averages and STDs of the synthetic ETref match the measured statistics for each calendar month. |
| ETDSAC | Correcting the daily ETref values of the ETDS time series such that the annual average and STD of the synthetic ETref match the measured statistics. |
| ETWD | The synthetic ETref values are randomly sampled from the empirical distribution of ETref values for rainy days or for dry days for each calendar month. |
| ETWDMC | Correcting the daily ETref values of the ETWD time series such that the monthly averages and STDs match the measured statistics for each calendar month. |
| ETWDAC | Correcting the daily ETref values of the ETWD time series such that the annual average and STD match the measured statistics. |

where $\mathrm{GR}(r,y)$ is the GR predicted by realization $r$ of the atmospheric conditions for year $y$, $N_y$ is the number of years for which the GR is predicted, $N_r$ is the number of atmospheric condition realizations, and the average is defined as:

$$\langle \mathrm{GR} \rangle = \frac{1}{N_r} \frac{1}{N_y} \sum_{y=1}^{N_y} \sum_{r=1}^{N_r} \mathrm{GR}(r,y). \tag{8}$$

It is easy to show that the total variability can be decomposed into the temporal and the realization contributions as (e.g. Strobach and Bel, 2017):

$\sigma_{total}^2 = \sigma_{temporal}^2 + \sigma_{realizations}^2,$ \hfill (9)





where,

$$\sigma_{temporal}^2 = \frac{1}{N_r}\sum_{r=1}^{N_r}\frac{1}{N_y}\sum_{y=1}^{N_y}\left(\text{GR}(r,y) - \langle\text{GR}\rangle_r\right)^2; \tag{10}$$

$$\langle\text{GR}\rangle_r = \frac{1}{N_y}\sum_{y=1}^{N_y}\text{GR}(r,y); \tag{11}$$

and

$$\sigma_{realizations}^2 = \frac{1}{N_r}\sum_{r=1}^{N_r}\left(\langle\text{GR}\rangle_r - \langle\text{GR}\rangle\right)^2. \tag{12}$$

This decomposition allows us to assess the number of realizations required to estimate the GR variability.

### 2.3 Model setup

The GR fluxes were calculated using the 1D Richards equation,

$$\frac{\partial\theta}{\partial t} = \frac{\partial}{\partial z}\left[K(\psi)\left(\frac{\partial\psi}{\partial z} + 1\right)\right], \tag{13}$$

where $\psi$ is the matric potential head $[L]$, $\theta$ is the volumetric water content (dimensionless), $t$ is time $[T]$, $z$ is the vertical coordinate $[L]$, and $K(\psi)$ $[L\ T^{-1}]$ is the unsaturated hydraulic conductivity function. The Richards equation was numerically simulated using the Hydrus 1D (Šimůnek et al., 2009). Atmospheric boundary conditions with surface runoff were prescribed at the upper boundary as precipitation, ETref (potential ET), and the minimum allowed pressure head at the soil surface

(hCritA$= -100000$ cm). Lower boundary conditions were prescribed as free drainage, and the length of the simulated soil column was 500 cm.

Knowledge regarding the soil hydraulic functions is essential in order to solve the Richards equation. The soil retention curves and the unsaturated hydraulic curves are commonly described according to the van Genuchten-Mualem (VGM) model (Mualem (1976); Van Genuchten (1980)):

$$S_e = \frac{\theta - \theta_r}{\theta_s - \theta_r} = \left[1 + (\alpha|\psi|)^n\right]^{-m}, \tag{14}$$

where $S_e$ is the degree of saturation ($0 < S_e < 1$), $\theta_s$ and $\theta_r$ are the saturated and residual volumetric soil water contents, respectively, and $\alpha$ $[L^{-1}]$, $n$, and $m = (1-1/n)$ are shape parameters. Hydraulic conductivity is assumed to behave according to:

$$K(S_e) = K_s S_e^l\left[1 - \left[1 - S_e^{1/m}\right]^m\right]^2 \tag{15}$$

where $K_s$ $[L\ T^{-1}]$ is the saturated hydraulic conductivity, and $l$ is the pore connectivity parameter prescribed as $0.5$.

The global distribution of soil texture provided by Hengl et al. (2014) and the aridity definition by the Food and Agriculture Organization (FAO, 2019) were used to determine the common soil types in arid and semi-arid environments (Figure 2). It was





found that 94% of the soil types in semi-arid and arid regions according to the USGS soil characterization are Clay Loam, Loam, Sandy Clay Loam and Sandy Loam (Figure 2). We used the soil hydraulic parameters provided by Carsel and Parrish (1988) that correspond to each of these soil types (Table 2). Note that only homogeneous soil profiles are simulated in the current study. We mention again that soil heterogeneity, local topography, vegetation and other complex soil water processes are likely to affect GR estimations. However, these additional complexities have to be analyzed separately and require detailed and location specific observations.

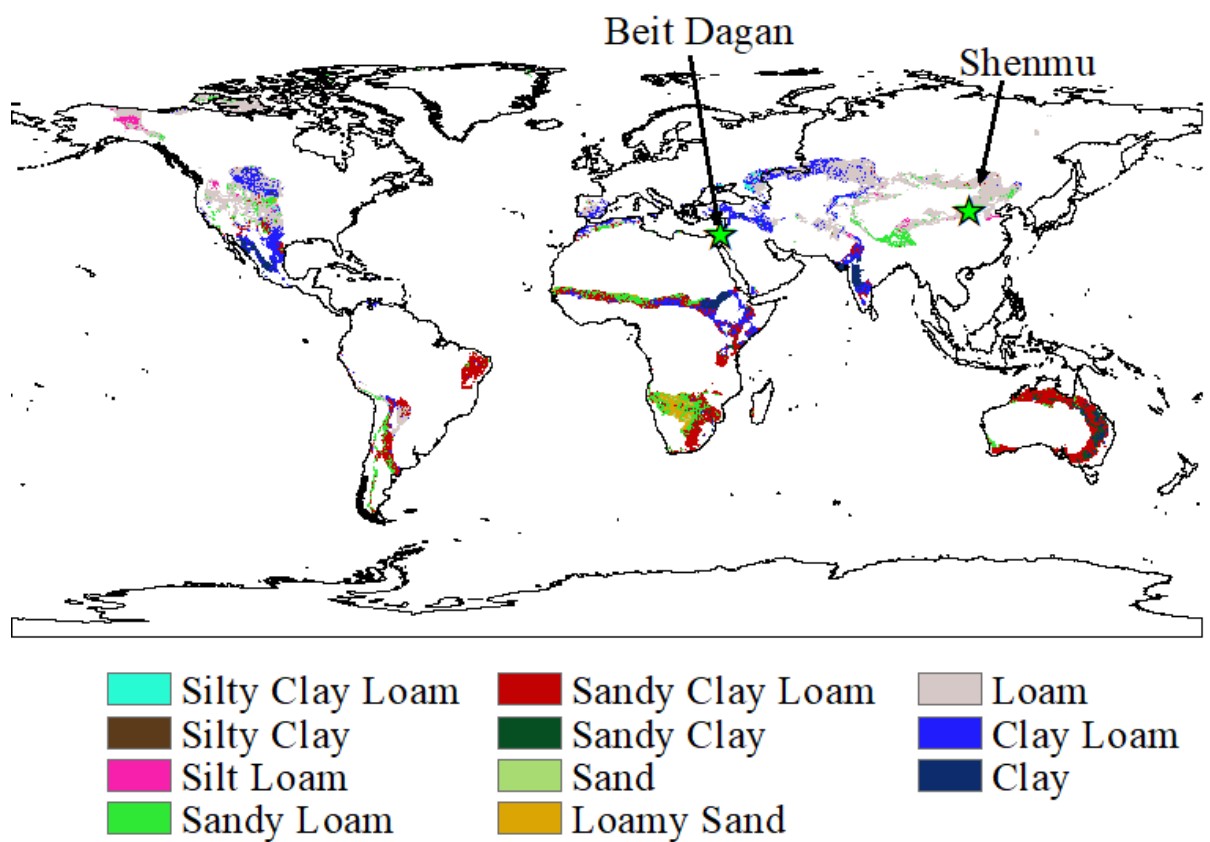

**Figure 2.** Global distribution of soil types in semi-arid and arid regions (the aridity index is between 0.2 and 0.5) according to the Food and Agriculture Organization (FAO, 2019) and the spatial soil texture distribution in these regions as provided by Hengl et al. (2014).

## 3 Results

### 3.1 Characteristics of the different methods to synthesize the rain and the ETref

Figure 3 shows the monthly rain statistics for each of the rain synthesis methods (DS, MCDS, ACDS, and FNRD). Note that comparisons between the measured and synthetic daily (Figures S1 and S2) and monthly (Figures S3 and S4) rain cumulative





**Table 2.** The soil hydraulic parameters used in the current study. The typical soil types in semi-arid and arid regions were identified according to the analysis presented in Figure 2. The parameters corresponding to each soil type were derived from the lookup table at Carsel and Parrish (1988).

| Soil type | $\theta_r$ | $\theta_s$ | $\alpha\ [cm^{-1}]$ | $n$ | $K_s\ [cm/day]$ |
|---|---|---|---|---|---|
| Clay Loam | 0.095 | 0.41 | 0.019 | 1.31 | 20 |
| Loam | 0.078 | 0.43 | 0.036 | 1.56 | 25 |
| Sandy Clay Loam | 0.1 | 0.39 | 0.059 | 1.48 | 31 |
| Sandy Loam | 0.057 | 0.41 | 0.124 | 2.28 | 350 |

distribution functions (CDFs) for each rainy month appear in the supporting information. Figures S5 and S6 in the supporting information provide a similar comparison between the distributions of the number of rainy days. In addition, Figures S7 and
S8 show the comparison between the measured and synthetic annual rain CDFs.

The DS method, which preserves the daily rain statistics and the statistics of the number of rainy days in each month, generates mean monthly rain amounts that are close to the observed values. However, both the monthly mean and monthly STDs ($\tilde{\sigma}_M(m)$) are generally underestimated. This occurs because the correlation between the number of rainy days and the total monthly rain was not accounted for. The FNRD method generates, in most cases, a higher mean monthly rain amount
but a lower STD than the DS method (Figure 3). Again, this is the result of not considering the correlation between the number of rainy days and the total rain amount. Namely, the number of rainy days is fixed, while the statistics of the daily rain amounts follows the empirical PDF extracted from the observations. It is apparent that the statistics of the MCDS method is the closest to the monthly observations, as expected (Figure 3). Figures S3 and S4 in the supporting information provide additional presentations of these characteristics by comparing the CDFs of the total monthly rain for each synthesis method
and the measurements.

The ACDS method tends to overestimate the mean and STD of the monthly rain during the most rainy months, while the monthly means are underestimated during the driest months (Figure 3). This is because the STD correction in this method, which is actually a multiplication by a factor larger than 1, amplifies the large rain events more than the small ones. Note that this correction is needed because the DS tends to underestimate the annual rain amount and STD. Figures S7 and S8 in the
supporting information provide additional presentations of these characteristics by comparing the CDFs of the total annual rain for each synthesis method and the measurements.

The monthly statistics of the synthesized ETref generated by the ETDS, ETWD, ETDSMC, ETDSAC, ETWDMC, and ETWDAC are depicted in Figure 4. The ETDS and ETWD show mean values that are similar to the observed ones. Both methods underestimate the STD of ETref (see Figure 4). This underestimation stems from the fact that the daily ETref values
for each month are derived from the empirical distribution, which is based on the entire dataset. Therefore, correlations within the month and the variability between years with higher ETref values and those with lower values are not accounted for. In other words, the values of the synthesized ETref in each month mix observed values within this calendar month from different years of the observations. Therefore, the difference between the synthesized mean ETref values for a specific calendar month





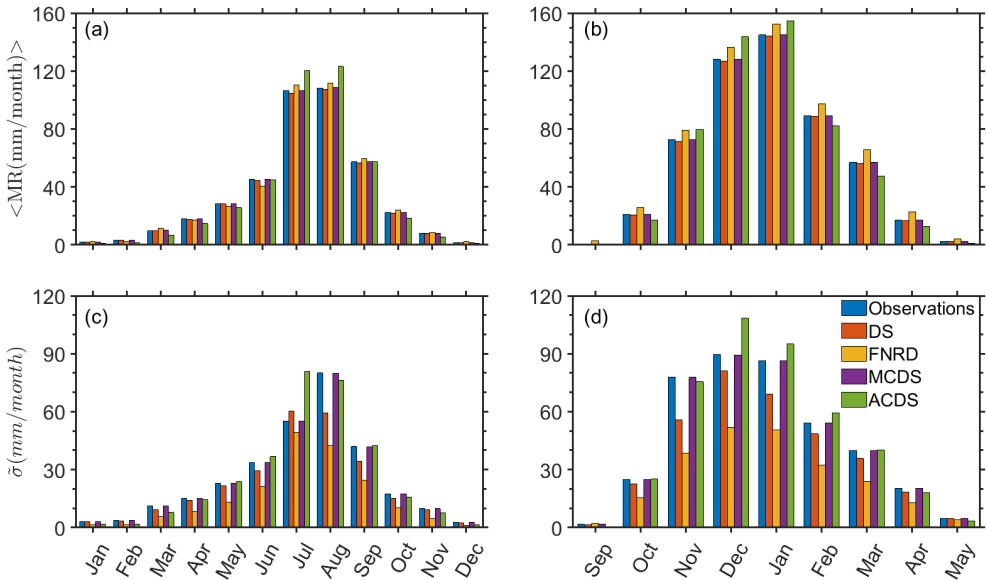

**Figure 3.** (a,b) Mean and (c,d) standard deviation of monthly rain for the (a,c) Shenmu climate and the (b,d) Beit Dagan climate. Note that the colors indicate the method of rain synthesis.

in different years is smaller than the observed difference. The similarity between the statistics of the two methods suggests that

the correlation between the ETref values and the rainy/dry nature of the day is not apparent. We also tested the importance of the ETref trend within the calendar month, and we found that it does not affect the observed statistics.

     Following the monthly corrections (equations (1)-(3)), the monthly mean and STD of the ETref that were synthesized by the ETDSMC and ETWDMC methods match the observed values (Figure 4). Both methods show very similar results for the mean and STD monthly values (see Figures S9–S14 of the supporting information for comparisons between the measured

and synthetic daily (Figures S9 and S10), monthly (Figures S11 and S12), and annual (Figures S13 and S14) ETref CDFs). This similarity suggests that the correlation between the ETref value and the rainy/dry nature of the day is not very strong. As expected, the mean and STD for both methods are very close to the observed monthly values.

     For both the ETDSAC and ETWDAC methods, higher monthly means were calculated during the warm months compared with the observations, while lower values were estimated for the cold months (Figure 4). Note that the number of rainy days

is synthesized for each month separately. Thus, the correlations between the number of rainy days in different months within the same calendar year are not accounted for. This, in turn, results in small differences between the synthesized ETDSAC and ETWDAC STD values.





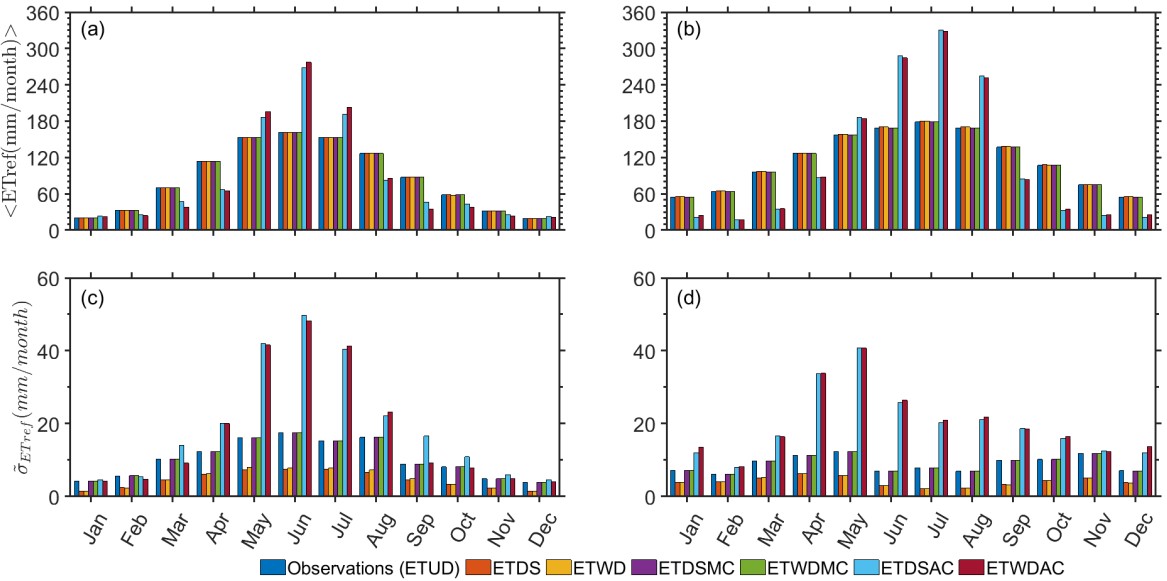

**Figure 4.** (a,b) Mean and (c,d) standard deviation of monthly ETref for the (a,c) Shenmu climate and the (b,d) Beit Dagan climate. Note that the colors indicate the method of ETref synthesis.

## 3.2 A comparison between estimated GR using observed and DS synthesized climate data series

To better understand the effects of the DS synthesis method we calculated (using the 1D Richards equation) the GR fluxes
under the observed and DS synthesized climate conditions. The calculations were done for both locations, Shenmu and Beit-Dagan. In addition, we used the CRU-ST reanalysis data and the corresponding DS synthetic data (the DS in this case was based on the ePDF of the reanalysis data) and repeated the GR flux calculations. The abovementioned estimated GR fluxes are depicted in Figure 5.

For all the soil types considered, higher annual average (and STD) GR fluxes are estimated when the observed climate data
in Shenmu is used (relative to the CRU data; Figure 5, four top panels). In Beit-Dagan there is no significant difference in the estimated annual average (and STD) GR fluxes between the observed and CRU data (Figure 5, four bottom panels). The estimated GR fluxes in Shenum (Figure 5, four top panels) resemble previously reported GR fluxes in the Loess Plateau. Tao et al. (2021) recently suggested that for loam and sandy clay loam textures ($\sim$20% clay), the GR fluxes range from 24.5 to 33.8 mm/yr. This values are similar to our results when the observed climate conditions are used and are more than the estimated GR
fluxes when the CRU data is used. Other studies suggested a range of 50 to 90 mm/year (Huang et al., 2017; Gates et al., 2011), which is closer to the estimated values in coarser soil textures (sandy loam). Regional analyses showed larger range of GR flux over the entire loess Platue between 0 and 75 mm/year (Hu et al., 2019; Turkeltaub et al., 2018), and Wu et al. (2019) reported GR fluxes larger than 100 mm/year for some specific location, similar to our estimated GR fluxes for sandy loam soil (Figure 5 top right panel). Our estimated GR fluxes for Beit-Dagan are also similar to those reported in previous studies. Eriksson





and Khunakasem (1969) reported GR fluxes that range between 30 and 326 mm/year along the coastal aquifer of Israel (our results are within this range and are in-line with the estimation for the specific soil types considered here). A commonly used recharge coefficient for the Israeli coastal aquifer is about 0.3 of the rain (Gvirtzman, 2002). Given that the average rainfall in the region is ~600 mm/year, the corresponding average GR flux is about 200 mm/year (similar to the our estimates for the finer soil types). Higher GR rates (330 mm/year) were estimated in a site near Beit Dagan, which is characterized by a sandy soil,

were reported by Turkeltaub et al. (2015), in agreement with our estimate for the sandy loam. These findings are supported by chloride and water isotope observations (Kass et al., 2005; Rimon et al., 2011). For heavy soils (e.g., clay loam) the range of the GR fluxes is wider and in many cases depends on local vegetation, agricultural activity and preferential flow through cracks (Kurtzman and Scanlon, 2011; Baram et al., 2012; Kurtzman et al., 2016). We emphasize again that the effects of these factors on GR flux estimations are beyond the scope of this study.

The GR fluxes that were calculated using the DS synthesized rain and ETref mostly show smaller means and STDs compared with GR fluxes estimated using the observed climate data (Figure 5). Similar trend is apparent also for the CRU and the corresponding DS synthesized data. This is an outcome of narrower yearly rain and ETref distributions for the DS synthesized variables (supporting information Figures S7, S8, S13 and S14). The DS method does not account for the correlation between the amount of rain and the number of rainy days or the existence of rainy (dry) years. Similarly, the ETDS method does

not account for the inter-annual variability, namely, the existence of warm and cold years. In both methods the statistics for each calendar month is based on the entire recorded time series and, therefore, the resulting distributions are narrower. This difference between the observed and the DS synthesized statistics inspires the implementation of various correction methods. However, the choice of the most adequate correction method is not straightforward and will be discussed later on.

### 3.3 GR Estimations using different corrections for the DS synthesis

Groundwater recharge sensitivity to different rain characteristics is examined by constraining the synthesized rain (DS) to yearly or monthly statistics as described in the Methods section. Additionally, the GR fluxes that were calculated when using only an average number of rainy days (FNRD) or when the monthly rain and ET are distributed over the month (UDDS) are compared with the DS method. The estimated GR fluxes are presented as the average annual GR fluxes together with their STDs $\sigma_{total}$ (Figure 6). Note that the symbols represent the soil types, and the colors indicate the ETref methods (Figure 6).

Thus, in total, we used 29 different methods to estimate the GR fluxes for each soil type and climate (Figure 6).

In general, under both climate conditions, the estimated GR increases in the following order: Clay Loam < Loam < Sandy Clay Loam < Sandy Loam (Figure 6). Furthermore, the estimated GR fluxes under winter rain are substantially higher than those under the summer rain climate (Figure 6). When applying the ACDS method, i.e., correcting the synthesized rain to match the recorded yearly statistics, the estimated GR fluxes are considerably higher than those estimated using the other rain

synthesis methods (Figure 6). Using the UDDS method, where both the monthly rain and the ETref are uniformly distributed over the month, resulted in considerably lower GR fluxes compared with the other methods. The estimated GR fluxes using the FNRD have similar mean values and higher STD values relative to the GR fluxes estimated using the DS method (Figure

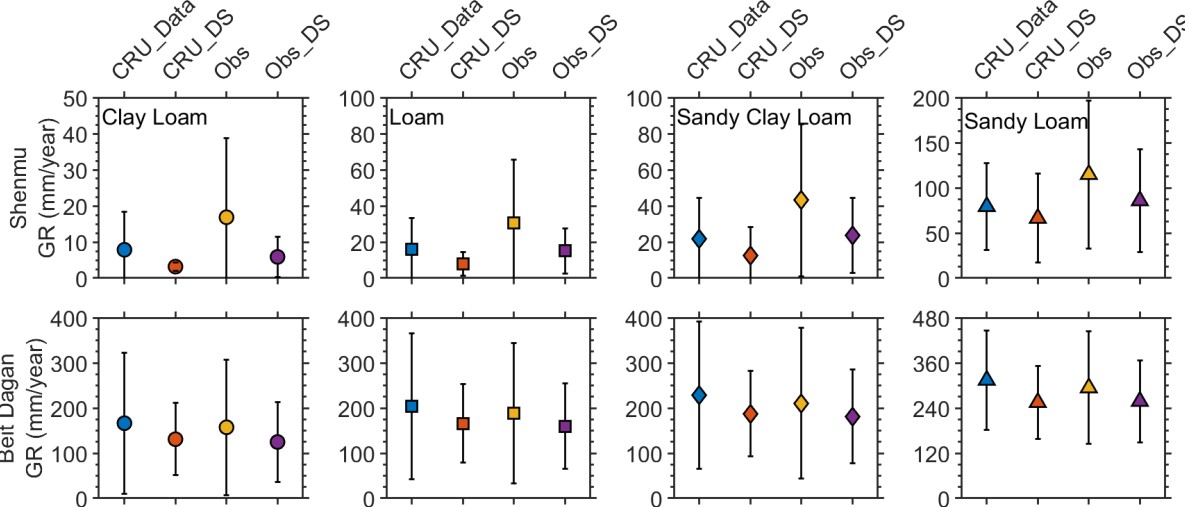

**Figure 5.** The four top panels present the Shenmu (summer rain) estimated GR fluxes and the four bottom panels present the Beit-Dagan (winter rain) estimated GR fluxes. Obs denotes the GR fluxes that were calculated using the observed rain and ETref; CRU_Data denotes the GR fluxes that were calculated using the downscaled CRU TS climate datasets (Harris et al., 2014; van Beek, 2008); DS_Obs denotes the GR fluxes that were calculated using the DS synthesized rain and ETref that were established according to the observed data statistics; CRU_DS denotes the GR fluxes that were calculated using the DS synthesized rain and ETref that were established according to CRU TS statistics.

6). Therefore, it appears that the distribution of the number of rainy days is not crucial when establishing a rain series for GR studies. The GR fluxes, estimated using the MCDS method, are slightly larger than those estimated using the DS method.

The ETref synthesis methods preserving the daily (ETDS and ETWD) or monthly (ETDSMC and ETWDMC) statistics and the uniformly distributed ETref (ETUD) result in similar GR fluxes. However, the ETref synthesis methods preserving the yearly statistics (ETDSAC and ETWDAC) result in considerably higher GR fluxes. This effect stems from the fact that these correction methods widen the distribution of the ETref values. However, while smaller values of ETref indeed decrease the actual evapotranspiration, the much larger values of ETref do not affect the actual evapotranspiration because it already

reaches an upper limit. The difference between the ETref values during wet and dry days is not affecting the GR fluxes. For all the correction methods, there were no apparent differences between the ETref synthesis methods that accounting for the wet/dry differences and those that do not.

     We further investigated whether the number of realizations (50 in the current study) is sufficient to adequately analyze the effect of the rain and ETref synthesis methods on GR fluxes. Both the $\sigma^2_{temporal}$ and $\sigma^2_{total}$ were calculated for 200, 20, and

10 years according to equation (9) (see also Figure 7). Calculating the ratio between the $\sigma^2_{temporal}$ and $\sigma^2_{total}$ illustrates that $\sigma^2_{temporal}$ constitutes most of the variability (Figure 7c,d). These results help confirm that increasing the number of realizations would have little effect on the estimated variance of the GR fluxes and that the number of realizations we used is sufficient.





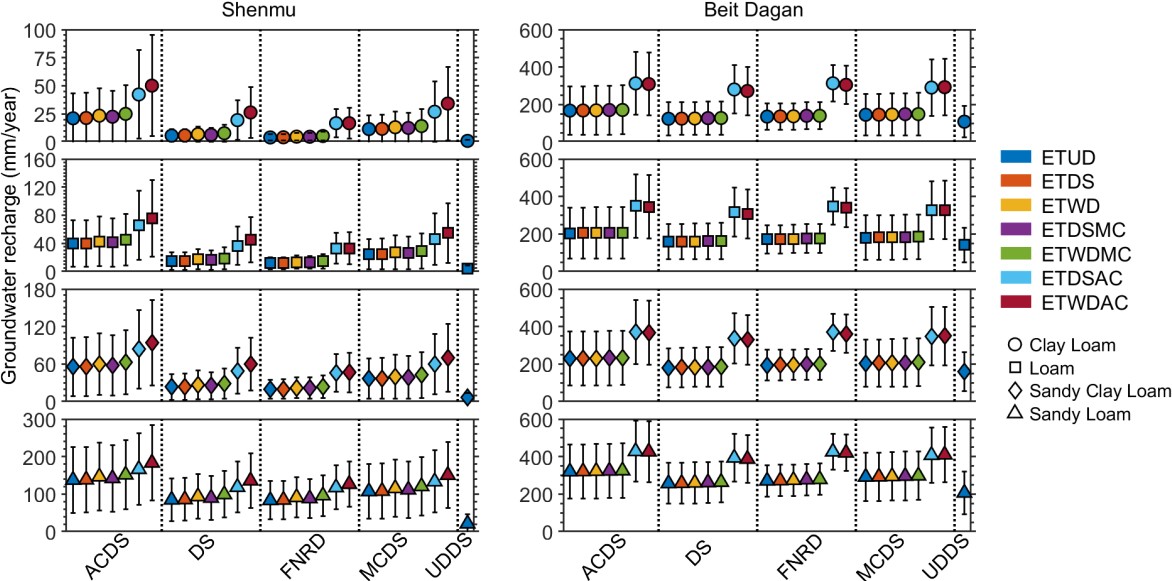

**Figure 6.** Estimated perennial GR based on 50 realizations for each of the typical soil types for the Shenmu and Beit Dagan climates. The different colors indicate the method of ETref synthesis. Note that the error bars represent the total STD, $\sigma_{total}$.

### 3.4 Correlation between annual rainfall and annual GR

The relationship between annual rain and annual GR fluxes are commonly used for regional GR estimations (e.g. Wohling et al.
(2012)) and to test possible future changes in GR fluxes (e.g. Crosbie et al. (2013)). Thus, the correlations between annual rain and annual GR fluxes for each combination of soil type, ET synthesis method, and rain synthesis method are examined (Figure 8). Note that due to the water travel time in the vadose zone, there could be a lag between the rain time and the water front arrival time to the depth considered as the groundwater level (i.e., the GR time). Thus, a cross-correlation analysis was conducted for the different rain methods, ET methods and soil types (Figure S15 and S16 in the supporting information). Figure 8 presents
the correlation coefficients or the lag time showing the strongest correlation. For Beit-Dagan we found that for all soil types and for all the synthesis methods the strongest correlation is obtained for zero lag. Namely, the travel time is shorter than a year. For Shenmu we found that the lag time yielding the strongest correlation varies between 0–2 years according to the soil types and the synthesis methods. In general, the correlation is higher in Beit-Dagan than in Shenmu due to the fact that in Beit-Dagan the rain is in the winter, when the ET is low, and in Shenmu there are significant summer rains.

We find that there are high correlations between the annual rain and the annual GR for sandy loam soil under both climate conditions and, except for the UDDS method, regardless of the rain and ET synthesis methods (see panels (d) and (h) of Figure 8). This high correlation is expected due to the high infiltration rate, which reduces the actual ET. For the UDDS, in sandy loam soil, we find that the correlation is smaller for an area with summer rains due to the effect of the high ET during the rainy period.

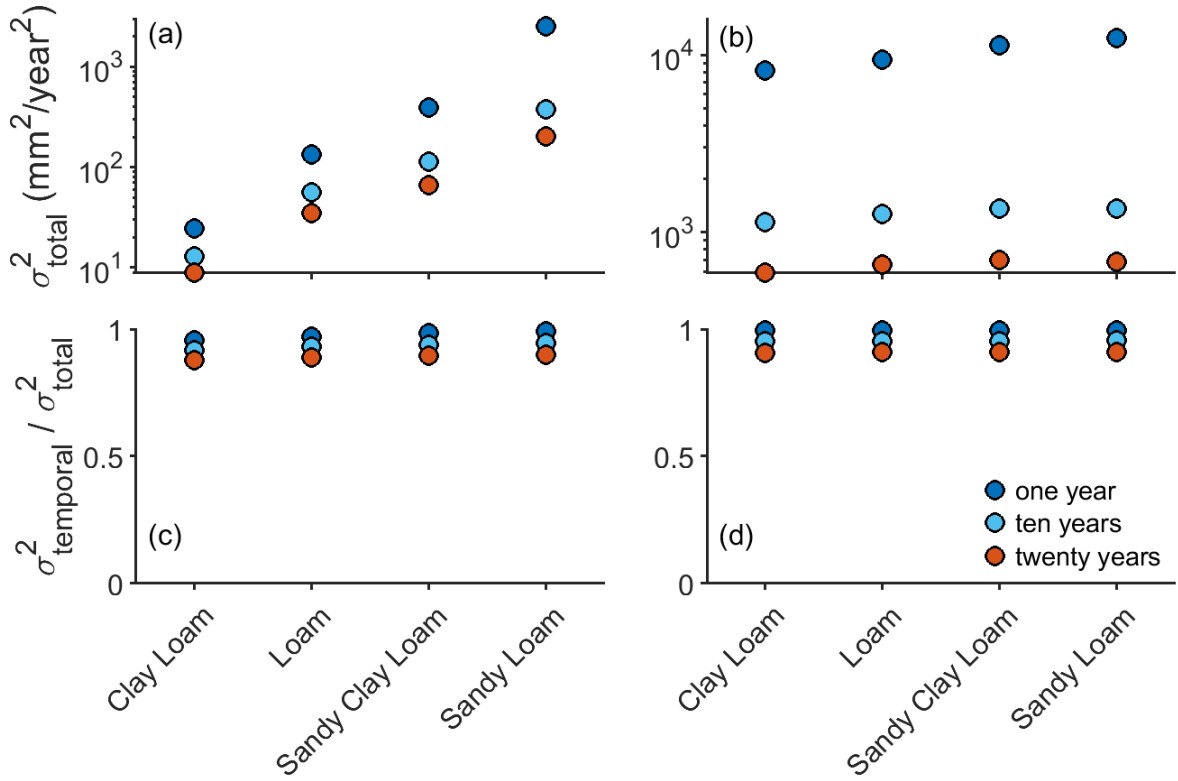

**Figure 7.** Total variance ($\sigma^2_{total}$) of the estimated perennial GR for (a) Shenmu and (b) Beit Dagan. The ratio between the internal variance ($\sigma^2_{temporal}$) and total variance ($\sigma^2_{total}$) is presented in the (a) Shenmu climate and the (B) Beit Dagan climate.

For the three other soil types, we find a considerably higher correlation under winter rain (panels ((a)–(c)) than under summer rain conditions (panels (e)–(g)). Under winter rain conditions, the correlation is high regardless of the ET and rain synthesis methods, with a somewhat weaker correlation for the UDDS method for the reasons mentioned above. The strongest correlation is obtained for the annually corrected ETref (for both the ETDSAC and ETWDAC). Under summer rain conditions, the ACDS rain synthesis method yields the highest correlation, and the FNRD yields the smallest correlation. Under summer rain, the

UDDS shows a negligible correlation, and the same is true for the FNRD with loam and clay loam soil types. Only the MCDS and ACDS yield apparent correlations due to the fact that these methods increase the frequency of large and small rain values.

### 3.5   The rain and GR ratio

The rain and ET synthesis methods affect the rain amount and the actual ET, respectively. Therefore, it is difficult to assess the effect of each method separately. In order to better delineate the role of the methods, we examine the fraction of rain that

turns into GR flux. In Figure 9, we depict the ratio between the accumulated rain and the accumulated GR flux. The left panels





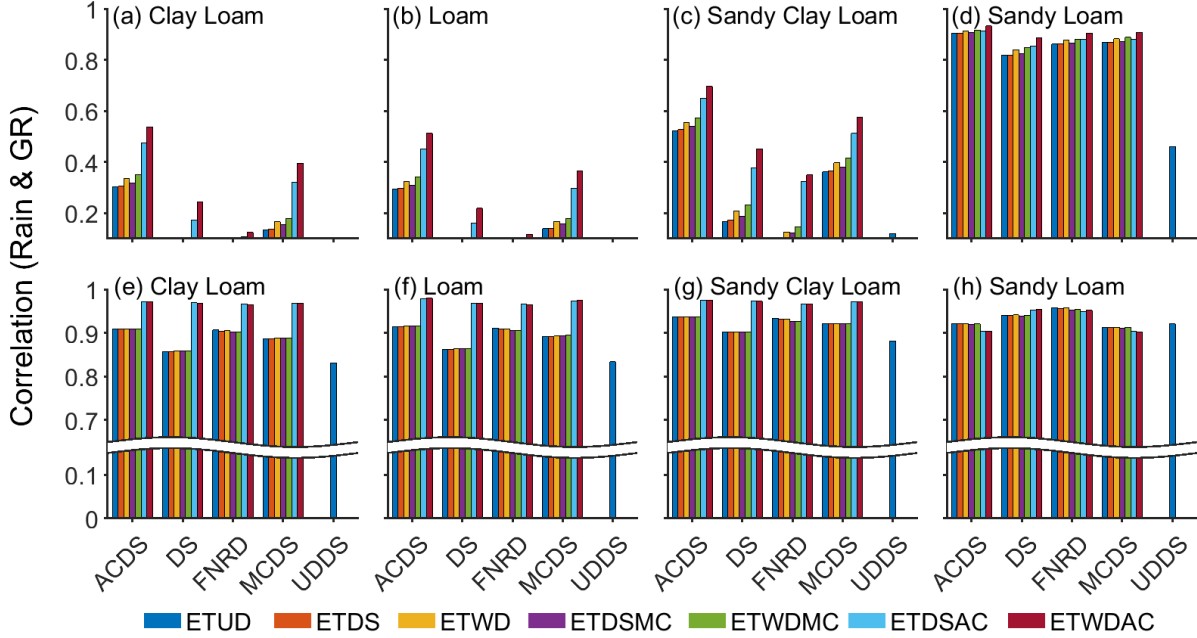

**Figure 8.** Correlation between annual rain and annual GR under the Shenmu climate, panels (a)-(d) and the Beit Dagan climate, panels (e)-(h), for (a and e) Clay Loam, (b and f) Loam, (c and g) Sandy Clay Loam, and (d and h) Sandy Loam. Note that the colors indicate the ET synthesis method.

correspond to Shenmu climate conditions, and the right panels correspond to Beit Dagan conditions. The rows correspond to the different soil types as indicated. In each panel, the ratio is presented for the different combinations of rain and ETref synthesis methods. There is similarity between the estimated GR/Rain ratio ranges (Figure 9) and corresponding ratios reported in the literature. Gates et al. (2011) indicated that the GR/Rain ratio is between 0.11 and 0.18 in the Loess Plateau, similar to

our estimates for the finer soil types under Shenmu climate conditions (panels a, c and e of Figure 9). For sandy soil, Zhang et al. (2020) reported GR/Rain ratio of about 0.58 in the Loess Plateau, slightly above our estimates for the sandy loam soil under Shenmu climate conditions (panel g of Figure 9). Gvirtzman (2002) suggested recharge coefficient of 0.3 for the entire Israeli coastal aquifer, which is lower than our estimates for the finer soil types under Beit-Dagan climate conditions (panels b, d and f of Figure 9). For sandy soil types, Eriksson and Khunakasem (1969) and Turkeltaub et al. (2015) showed that the GR

fluxes can be higher than 300 mm/year, corresponding to GR/Rain ratio larger than 0.5.

For winter rain, in Beit Dagan (right column of Figure 9), we find that the ratio is not very sensitive to the rain synthesis method, with the exception of the UDDS, which results in a lower ratio due to the expected higher actual ET. We also find that the ETDSAC and ETWDAC ETref synthesis methods yield the largest ratio, namely the largest fraction of rain infiltration to the groundwater. This higher ratio is the result of the wider distribution of ETref values, which results in smaller actual ET

(because ETref values above a certain threshold do not increase the actual ET).





For summer rain, the ratio is much smaller due to the larger ET. However, the same dependence on the rain and ETref synthesis methods that is observed for Beit Dagan is also apparent here.

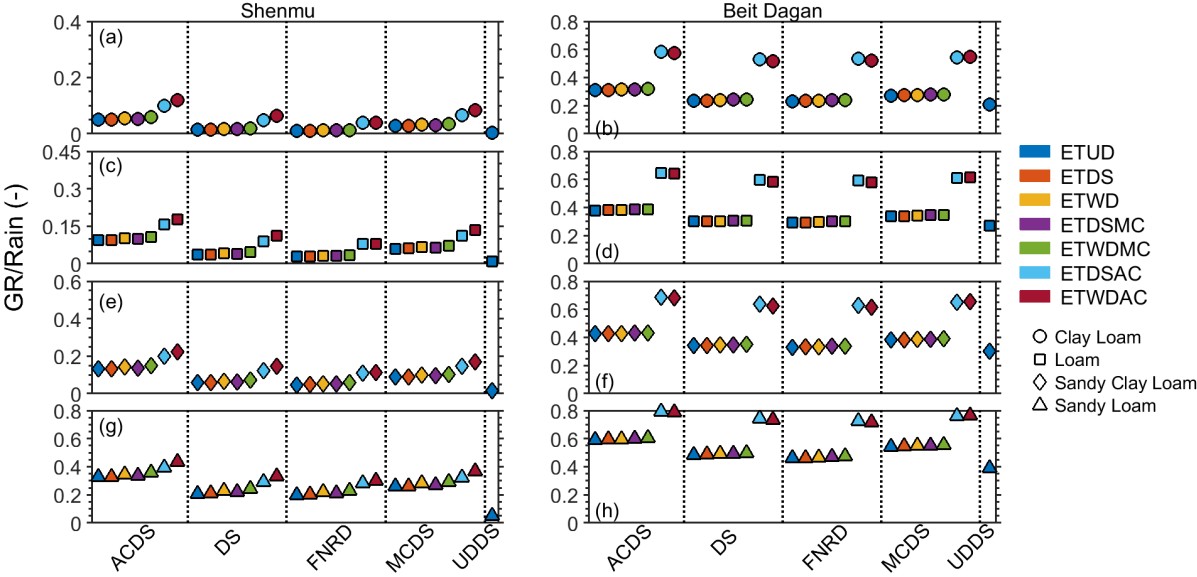

**Figure 9.** The ratio between GR and rain for different climate conditions, rain and ETref synthesis methods, and soil types.

## 4   Conclusions

There are different methods that can be used to synthesize the rain and ETref in order to study the dynamics of GR fluxes.

The synthesis is often based on measurement records, but only certain aspects of the measured statistics are preserved in the synthesis method. Here, we considered five different methods for rain synthesis and seven methods for ETref synthesis. The different methods are all based on the statistics of the measured daily values for each calendar month. However, preserving the daily statistics alters the monthly and annual statistics, and the methods considered here offer corrections in order to match the mean and variance of the monthly or annual statistics. We find that the statistics of GR fluxes depends on the synthesis method.

Namely, the GR fluxes depend on the characteristics of the rain and ETref statistics that were preserved in the synthesis method. For winter rain conditions, the different synthesis methods yield similar GR fluxes and similar ratios between the GR fluxes and the rain amounts. Notably, preserving the annual statistics of ETref yields substantially higher GR fluxes and higher ratios between the GR flux and the rain amount. This effect is due to the broader distribution of ETref values that this method yields. The very large ETref values do not increase the actual ET, while the low values decrease the actual ET, thereby increasing

the GR flux. For summer rain conditions, the fraction of rain that infiltrates is smaller than in winter rain conditions due to the increased ET. However, a similar dependence on the synthesis methods is found. An exception to the abovementioned remarks is the GR flux through sandy loam soil, which shows a much weaker sensitivity to the synthesis methods and climate





conditions, due the fast infiltration rates of surface water in this soil type. The correlations between the annual rain and the GR flux vary with the climate conditions, soil type, and rain and ETref synthesis methods. Therefore, one should be careful when
applying measurement-based statistics of rain and ETref to studies of GR fluxes. The most representative synthesis method may be very different for different locations, with certain soil types and climate conditions.

The distribution of the annual GR estimated by the monthly corrected DS data is found to be the closest to the distribution of the estimated annual GR using the observed data. Yet, this conclusion is limited to the two locations considered here and to the simplified model used. Further studies spanning more climate conditions, soil types and soil water models are required in
order to explain and generalize this result.

*Code availability.* The code used in this research will be made available upon a reasonable request.

*Data availability.* The data used in this research will be made available upon a reasonable request.

*Author contributions.* TT and GB designed the research, analyzed the data, and wrote the manuscript.TT performed the numerical simulations.

*Competing interests.* The authors declare no competing interests.



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
