# Peer review of "The effects of rain and evapotranspiration statistics on groundwater recharge estimations under semi-arid environments"

_Hydrology and Earth System Sciences, 2022_

## Referee Comment (RC2)

Manuscrip ID: hess-2022-257

Comments and suggestion for authors

General comments

This study identified the important characteristics of the local rain and ETref for estimating the diffuse recharge under semi-arid climate conditions. I consider that the evaluation of the synthesis methods for rain and ETref is a great contribution since these variables have great importance on GR models. My only concern is the assumption of a homogeneous soil and only two locations considered, but they bear in mind this situation. However, some corrections are needed to improve the quality of the manuscript.

1. The abstract is too popular and not scientific enough to reflect the conclusion of the manuscript. Some sentences should be rewritten and supported by data.
2. In the introduction, the necessity of this study must be stated more explicitly.
3. In conclusion, it is necessary to reflect what is the scientific novelty of the results of your research. On the other hand, which synthesis method performed better for both rainfall and ETref?
4. What is the importance of understanding rainfall and evapotranspiration characteristics on groundwater recharge?

Specific comments

In line 1, you should eliminate the word *climate* in order to be clearer, since only the factor rainfall is boarded.

In lines 37-38, this sentence should be added to the next paragraph. Since I understand that paragraph explains the Poison approach.

In lines 59-61, *"To 60 overcome this issue, several new methods, preserving different characteristics of the measured rainfall and ETref records, are applied"*, you must be more specific, which methods did you employ for rainfall and ETref?

In lines 156-158, *"the two- sample Kolmogorov-Smirnov test indicated that the synthesized and the measured distributions are statistically similar for the DS, ETDS, and ETWD methods"*, which is the value of the KS test for this affirmation?

In line 325, *"We find that there are high correlations between the annual rain and the annual GR for sandy…"*. Please specify which correlation criteria is used, it means which values correspond to high correlations. Also add these values to the sentences. Please, review this in the manuscript.

In line 361, *"Here, we considered five different methods for rain synthesis and seven methods for ETref synthesis"*. In total you applied 12 synthesis methods, however, in Table 1 only appear 11. Please correct this.

Technical corrections

In subsection 2.2 Generation of rain and ETref time series. Consider separate rain and ETref in two subsections (e.g., 2.2. and 2.3) to facilitate the reading.

In line 50, you employed ETref for potential evapotranspiration. However, in line 71 you refer for reference evapotranspiration. Take this into account and make the corresponding corrections throughout the manuscript.

In Figure 5, I suggest the use of letters (A,B,C,…) such as in Figure 8, to facilitate the reading and location of the graphic. For example, in lines 263-265 "Figure 5 top right panel" could be hard to follow.

---

## Author Response (AR1)

Dear Editor,

We thank you and the reviewers for the careful assessment of our manuscript and for the constructive criticism. Below we address the reviewers' comments one by one. For clarity, we use regular black font for the quoted reviewer comments and blue italicized font for our responses.

**Reviewer 1**

It is common to synthetically generate essential data series including rain and evaporation based on limited station record in practice. This paper aimed at the impact of how these synthetic series are generated/corrected and investigated several methods such as fixed number of rainy days, monthly averaging, annual averaging etc (five synthesis methods for rain and seven for ETref). These synthetic data are then used in predicting groundwater recharge in one-dimensional Hydrus models with different soil types, and factors influencing groundwater recharges including methods to synthesize data as well as climate and soil type, are analyzed. The results indicated that groundwater recharge statistics indeed depend on data statistics perserved in synthesis methods. It brings attention to the synthesis methods to generate rain and ETref in groundwater applications. I do believe that the conclusions in this paper are very important in a broad community.

However, I agree that the authors are onto a good idea and the manuscript is well written, clear and detailed.

My only major concern is that: this study bears a significant limitation of considering only perfectly uniform soil, which greatly constrains the potential values of this study, as is also stated by the authors. If not so, I'd recommend

acceptance of this manuscript without hesitation. Hence, it is highly suggested that the authors take factors such as topography and soil structure into account in their ongoing work.

**Reply to general comment 1:** *Indeed, the work is limited to homogeneous soil, and the important effects of heterogeneity are not accounted for. However, as you mentioned, this work focus on the effects of the climate conditions synthesis methods on the simulated groundwater recharge. We felt that adding the complexity of heterogeneity would overload the MS. In future studies, we will address the effects of heterogeneity in similar scenarios and in multiple semi-arid locations worldwide.*

**Specific comments:** Lines 28-29: Gurdak et al. 2007 apeared twice. Are these two references the same?

*Thank you for finding our typo with the reference in lines 28-29. We corrected it in the revised MS.*

**Reviewer 2**

**General comments**

**General comment 1:** The abstract is too popular and not scientific enough to reflect the conclusion of the manuscript. Some sentences should be rewritten and supported by data.

**Reply to general comment 1:***We revised the abstract and added, in multiple places, specific results and data values in relevant sentences. Specifically, the revisions appear in lines 5-7,9-10 and 14-16.*

**General comment 2:** In the introduction, the necessity of this study must be stated more explicitly.

**Reply to general comment 2:** *We added text to better clarify and emphasize the goals of the current study. In lines 59-60, we added the following text: "Specifically, we test the sensitivity of estimated GR fluxes to the daily, monthly, and annual statistics of rain and ETref."*

**General comment 3:** In conclusion, it is necessary to reflect what is the scientific novelty of the results of your research. On the other hand, which synthesis method performed better for both rainfall and ETref?

**Reply to general comment 3:** *The text in the Introduction and Conclusion sections was revised to provide the missing information (lines 19-21 and 380-382).*

**General comment 4:** What is the importance of understanding rainfall and evapotranspiration characteristics on groundwater recharge?

**Reply to general comment 4:** *Understanding the effects of rainfall and evapotranspiration characteristics on groundwater recharge is important for both fundamental understanding of the underlying physical processes and for groundwater management under future climate conditions. The text was revised to include this information (lines 19-21).*

**Specific comments**

**Comment 1:** In line 1, you should eliminate the word climate in order to be clearer, since only the factor rainfall is boarded.

**Reply Comment 1:** *The word has been removed as suggested.*

**Comment 2:** In lines 37-38, this sentence should be added to the next paragraph. Since I understand that paragraph explains the Poison approach.

**Reply Comment 2:** *The suggestion was implemented in the revised MS (line 36).*

**Comment 3:** In lines 59-61, "To 60 overcome this issue, several new methods, preserving different characteristics of the measured rainfall and ETref records, are applied", you must be more specific, which methods did you employ for rainfall and ETref?

**Reply Comment 3:** *The full details of all the methods implemented are provided in the Methods section and throughout the MS. It would be impossible to provide the full details in the Introduction. The text has been revised in order to refer the reader to the section with the full details (lines 59-60, 63).*

**Comment 4:** In lines 156-158, "the two- sample Kolmogorov-Smirnov test indicated that the synthesized and the measured distributions are statistically similar for the DS, ETDS, and ETWD methods", which is the value of the KS test for this affirmation?

**Reply Comment 4:** *We now provide Table S1, in the Supplementary Information, that details the p-values of the KS tests for each method and location (line 162).*

**Comment 5:** In line 325, "We find that there are high correlations between the annual rain and the annual GR for sandy...". Please specify which correlation

criteria is used, it means which values correspond to high correlations. Also add these values to the sentences. Please, review this in the manuscript.

**Reply Comment 5:** *The text has been revised to provide the correlation coefficient values (lines 329-330). In addition, we wish to mention that these details appear in Figures S15 and S16 of the Supplementary Information.*

**Comment 6:** In line 361, "Here, we considered five different methods for rain synthesis and seven methods for ETref synthesis". In total you applied 12 synthesis methods, however, in Table 1 only appear 11. Please correct this.

**Reply Comment 6:** *The UDDS method specifies the synthesis methods of both the rain and the ETref (both are uniformly distributed over the month). This information appears in Table 1, and we added text to emphasize this (lines 365-366).*

**Comment 7:** In subsection 2.2 Generation of rain and ETref time series. Consider separate rain and ETref in two subsections (e.g., 2.2. and 2.3) to facilitate the reading.

**Reply Comment 7:** *Since the rain and ETref time series are generated by similar synthesis methods, we would prefer to preserve the original structure of the Methods section.*

**Comment 8:** In line 50, you employed ETref for potential evapotranspiration. However, in line 71 you refer for reference evapotranspiration. Take this into account and make the corresponding corrections throughout the manuscript

**Reply Comment 8:** *Thanks for catching this. The text has been revised (lines 51-52), and the term ETref now denotes the potential evapotranspiration*

*throughout the revised manuscript.*

**Comment 9:** In Figure 5, I suggest the use of letters (A,B,C,...) such as in Figure 8, to facilitate the reading and location of the graphic. For example, in lines 263-265 "Figure 5 top right panel" could be hard to follow.

**Reply Comment 9:** *Figure 5 has been revised to include panel labels, and the text was revised to refer to these labels.*